# Kinetically-controlled intermediate-direct-pinning for homogeneous energy landscapes in quasi-two-dimensional perovskites for efficient and narrow blue emission

Joo Sung Kim [1,2,3,9], Hyeon-Dong Lee[1,9], Jeong Hyun Jung [1,9], Taehee Kim [4,9], Seung-Je Woo [1], Hyung Joong Yoon[5], Jaehyun Moon[6], Chan-mo Kang [6], Seung-Eui Chang [1], Dong-Hyeok Kim[1], Sungjin Kim [1], Hoichang Yang[7], Dongho Kim [4] ✉ & Tae-Woo Lee [1,2,3,8] ✉

Quasi-two-dimensional perovskite structures hold great potential as active layers in blue perovskite light-emitting diodes. However, they face challenges of limited emission efficiency and broadened spectra due to phase inhomogeneity. Here, we report an intermediate-direct-pinning method to develop a uniform-phase quasi-two-dimensional structure with a homogeneous energy landscape. By forming a strong cation-π interaction complex, we stabilize a metastable intermediate phase with retarded crystallization toward low-$n$ phases ($n \leq 3$), followed by a direct pinning process favouring crystallization of medium-$n$ phases ($n = 4$ and 5) without further broadening. Additionally, introducing a surface-anchoring ligand during the pinning process effectively suppresses non-radiative recombination. The resultant structure shows efficient sky-blue emission and narrow linewidth (108 meV). Devices fabricated with this structure reach a maximum external quantum efficiency of 22.5% at 489 nm, which is scalable to large-area (pixel size: 900 mm²) and passive matrix devices (30 × 10 arrays, active area = 200 μm × 600 μm). These findings highlight the potential of perovskite light-emitting diodes for full-colour displays.

Metal halide perovskite light-emitting diodes (PeLEDs) exhibit outstanding optoelectronic characteristics such as high photoluminescence quantum yield (PLQY), efficient charge transport, and spectrally narrow emission for vivid displays and optical sources[1,2]. Recent advances have enabled PeLEDs to achieve external quantum

efficiency (EQE) > 25% in green and red emission[3–5], but the development of narrow-emitting blue PeLEDs with high efficiency remains a challenge[6–10].

Efforts to address this challenge involve the development of blue perovskite emitters through the pursuit of bandgap tunability in

[1]Department of Materials Science and Engineering, Seoul National University, 1 Gwanak-ro Gwanak-gu, Seoul, Republic of Korea. [2]Soft Foundry, Seoul National University, 1 Gwanak-ro, Gwanak-gu, Seoul, Republic of Korea. [3]SN DISPLAY Co. Ltd., Building 33, 1 Gwanak-ro, Gwanak-gu, Seoul, Republic of Korea. [4]Department of Chemistry, Yonsei University, 50 Yonsei-ro, Seodaemun-gu, Seoul, Republic of Korea. [5]Research Center for Materials Analysis, Korea Basic Science Institute (KBSI), 169-148 Gwahak-ro, Yuseong-gu, Daejeon, Republic of Korea. [6]Reality Devices Research Division, Electronics and Telecommunications Research Institute, 218, Gajeong-ro, Yuseong-gu, Daejeon, Republic of Korea. [7]Department of Chemical Engineering, Inha University, 100 Inha-ro, Michuhol-gu, Incheon, Republic of Korea. [8]Interdisciplinary Program in Bioengineering, Institute of Engineering Research, Research Institute of Advanced Materials, Seoul National University, 1 Gwanak-ro, Gwanak-gu, Seoul, Republic of Korea. [9]These authors contributed equally: Joo Sung Kim, Hyeon-Dong Lee, Jeong Hyun Jung, and Taehee Kim. ✉e-mail: dongho@yonsei.ac.kr; twlees@snu.ac.kr

mixed-halide composite structures. The Cl-Br mixed-halide composite emits blue with peak wavelength ($\lambda$) < 500 nm with an enlarged bandgap[11,12]. Unfortunately, facile occurrence of phase segregation between Br-rich and Cl-rich phases, exacerbated by an electric field, results in a gradual spectral shift and broadening of emission spectra.

Instead, quasi-two-dimensional (quasi-2D) structures have been used to extend the bandgap into the blue-emission region by exploiting the quantum confinement effect[6,13,14]. The quasi-2D structures have been shown to be promising for efficient blue emission with enlarged bandgap and strong charge confinement effect, achieving the high efficiency of blue PeLEDs with EQE up to 21.4% at $\lambda$ = 483 nm[10]. However, the quasi-2D structures featuring mixed quantum wells with different $n$ monolayers present challenges, such as inhomogeneous distribution of the $n$-phases. This distribution leads to additional broadening in energy distribution, thus the emission spectra broaden and a significant portion of charge carriers is lost during the charge-transfer process[15,16]. From this, most high-efficiency (EQE > 15%) blue PeLEDs based on low-dimensional structures suffer from broad electroluminescence (EL) linewidth ( > 120 meV) from the broad energy distribution[6–10,17]. To avoid these disadvantages, it is imperative to develop a perovskite material system with a homogeneous quantum-confined structure for efficient and narrow blue emission.

In this work, we develop a homogeneously-anchored quasi-2D structure with uniform $n$ monolayers by introducing the anchoring-assisted intermediate-direct-pinning (A-IDP) method. Instead of the rapid crystallization route toward low-$n$ phases ($n \leq 3$), we achieve a prolonged metastable state by inducing a strong cation-$\pi$ interaction with phenethylammonium (PEA$^+$) cation to form an intermediate complex. Subsequently, a direct pinning process is applied to selectively retain only medium-$n$ phases ($n$ = 4, 5). These procedures slow down the thermodynamic broadening and pin the phase distribution and ultimately yield homogeneous phase distribution with a narrow emission spectrum.

Further, the PEA$^+$ molecules at the crystal surface are substituted with a surface anchor molecule, 1,2-Cis-cyclohexyl dicarboxylic acid (CCA), which provides a stable crystal surface with reduced trap density. The perovskite structures obtained through the A-IDP method exhibit uniform nanosized structure with dominant medium-$n$ phases ($n$ = 4, 5), featuring fast charge funnelling and efficient light emission with narrow EL linewidth (108 meV). As a result, the blue PeLEDs with A-IDP process exhibit the EQE of 22.5% at $\lambda$ = 489 nm without an out-coupling structure, which is further enhanced to 41.8% by using a hemispherical lens for outcoupling. Also, we demonstrate uniform large-area blue PeLEDs with micropattern array (30 × 10 arrays, active area = 200 μm × 600 μm), showing the feasibility of PeLEDs for use in full-color display application.

## Results

### Anchoring-assisted intermediate direct pinning process

The broad phase distribution of quasi-2D perovskites is mainly attributed to the rapid crystallization of low-$n$ phases ($n \leq 3$) that have high formation energy[18]. Further, the thermal annealing treatment to crystallize perovskite can induce further broadening of phase distribution by promoting ion diffusion toward a thermodynamically stable distribution[16]. To inhibit this broadening effect, we introduced an additional intermediate state, then induced direct pinning of the homogeneous intermediate phases into uniform quasi-2D crystals that have a homogeneous energy landscape (Fig. 1a).

The intermediate phase was first induced by exploiting a strong cation-$\pi$ interaction between PEA$^+$ and ammonium (NH$_4^+$) ions that retards crystallization toward low-$n$ phases. Before thermodynamically driven crystallization of low-$n$ phases occurs, these intermediate phases can exist uniformly in a metastable state without the broadening of phases that occurs during one-step precursor-to-crystal conversion. Here, tetrahydrofuran (THF) as a volatile Lewis-base solvent and CCA

as an anchoring molecule were applied to the intermediate film to achieve an anchoring-assisted intermediate direct pinning method (A-IDP) (Supplementary Fig. 1). The CCA-THF solution was applied to the intermediate film and immediately spin-dried (hereafter referred as exposure duration, $D_E$ = 0 s); this process substitutes the CCA ligand for the intermediate complex of PEA, simultaneously removing residual solvents to prohibit diffusion-induced broad phase distribution of quasi-2D crystals. We performed the in-situ UV-Vis absorption analysis during the crystallization to monitor the kinetic changes in phase distribution via the A-IDP process (Supplementary Fig. 2). The pristine film without A-IDP process exhibited slowly increasing absorbance corresponding to the $n$ = 1, 2, 3 phases with each different rate, suggesting the intermixing between low-$n$ phases during the crystallization process. In contrast, the A-IDP film maintained the initial relative phase distribution throughout the whole crystallization process; this suggests the A-IDP process can kinetically fix the phase distribution and inhibit intermixing among the low-$n$ phases.

Grazing incidence wide-angle X-ray scattering (GIWAXS) patterns provided the crystal structure and orientation of the perovskite after A-IDP (Fig. 1b, c). The pristine perovskite film had highly-ordered diffraction peaks that correspond to the orientation of low-$n$ phases ($n$ = 1–3) toward the out-of-plane ($Q_z$) direction; this result suggests the spontaneous aggregation of perovskite crystals into low-$n$ phases during the crystallization process. The 1D XRD pattern showed intense diffraction peaks of low-dimensional phases oriented toward the $Q_z$ direction, which is an unfavorable configuration for charge transport toward the direction of charge injection during device operation[19].

In contrast, perovskite film prepared by A-IDP showed only ring-patterned scattering peaks, which correspond to 3D CsPbBr$_3$ (5.8 Å) oriented mainly toward the $Q_z$ direction, indicating increased crystallinity of medium-$n$ phases toward the vertical direction that can benefit on efficient charge transport in LED devices. Importantly, there were no diffraction peaks at low angle <1.0 Å$^{-1}$, indicating the removal of highly-ordered low-$n$ phases. A-IDP significantly reduced the surface root-mean-square (RMS) from 5.160 nm to 0.808 nm (Supplementary Fig. 3); this change indicates redistribution of aggregated low-$n$ phases into uniformly distributed nanosized medium-$n$ phases. Increasing $D_E$ up to 20 s intensified the diffraction peak of 3D phases in GIWAXS (Supplementary Fig. 4). However, complete removal of quasi-2D phases created pinholes at grain boundaries; to avoid this, the selective removal of excess intermediate phases requires precise control over short reaction times (Supplementary Fig. 5).

We examined the incorporation of CCA molecule into quasi-2D crystal structure by comparing pure $n$ = 1 PEA$_2$PbBr$_4$ crystals prepared by solvent-IDP (S-IDP) process with only THF solvent, with those prepared via the A-IDP process with CCA-THF solution. The A-IDP process applied to pure $n$ = 1 PEA$_2$PbBr$_4$ crystals led to a reduction in lattice distance from 16.72 Å to 16.63 Å, due to the smaller molecular size of CCA compared to that of PEA and strong coordination between PbBr$_6^{4-}$ octahedra and the −COOH and −C = O groups in CCA (Fig. 1d). Fourier-transform infrared (FTIR) analysis confirmed the removal of PEA after S-IDP process and incorporation of the CCA molecule after the A-IDP process (Supplementary Fig. 6a–c).

To further identify the chemical composite after the A-IDP process, XPS analysis was performed on samples with $D_E$ ranging from 0 s (A-IDP) to 20 s (Supplementary Fig. 6d–i). The core level peaks of C 1s and N 1s originating from PEA cations gradually diminished as $D_E$ increased, consistent with the removal of low-$n$ phases observed through GIWAXS. A comparison of the N/Pb atomic ratio revealed a stoichiometric shift from average value of <$n$> = 3 (N/Pb = 1.25) toward <$n$> = 5 (N/Pb = 1.0) after $D_E$ = 0 s, which eventually became a 3D-like composite with N/Pb ratio <0.4 as $D_E$ increased (Fig. 1e). The peak position and relative stoichiometry of Pb 4f and Br 3d peak were retained; this observation indicates selective removal of metastable

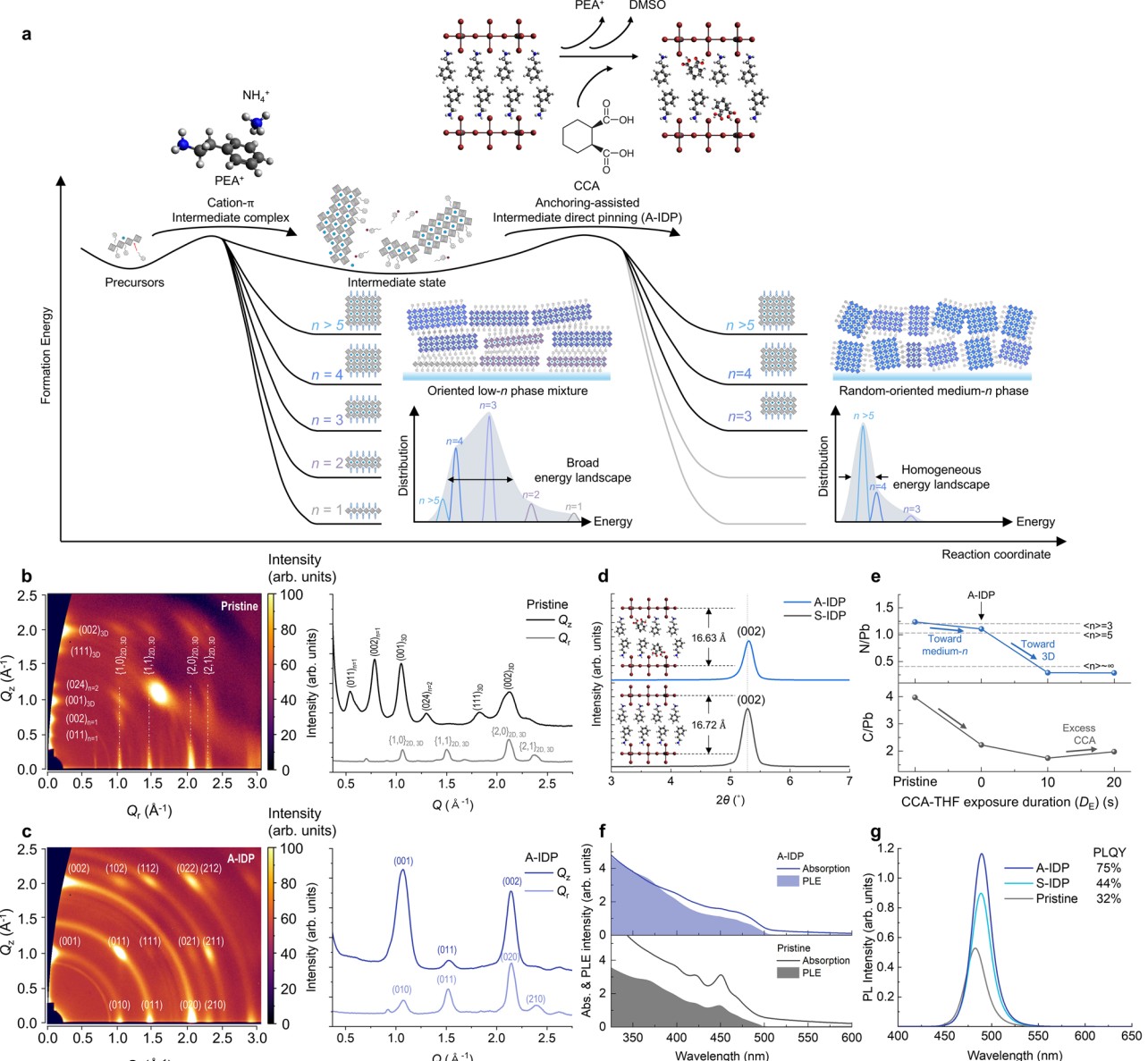

**Fig. 1 | Anchoring-assisted intermediate direct pinning for quasi-2D perovskites. a** Schematic reaction coordinate of the intermediate direct pinning (IDP) process of quasi-2D perovskites. **b, c** Grazing-incident x-ray diffraction pattern corresponding 1D XRD profile of pristine (**b**) and A-IDP (**c**) perovskite thin films. **d** X-ray diffraction spectra of $n = 1$ PEA$_2$PbBr$_4$ perovskites without and with CCA additives as molecular anchor. **e** Relative content of carbon and nitrogen to lead in perovskite films determined from XPS. **f** Normalized UV-vis absorption and PLE spectra of pristine and A-IDP perovskite films. **g** Steady-state PL spectra of perovskite films.

PEA cations without any compositional change in the perovskite crystal.

The influence of phase distribution on the behavior of charge carriers was elucidated through UV-vis absorption and photoluminescence excitation (PLE) spectra (Fig. 1f). The pristine film exhibited a distinct excitonic peak feature associated with low-$n$ phases (~425 nm and ~450 nm) situated well above the onset of absorption (~490 nm). In contrast, this feature was markedly subdued in the A-IDP film, indicative of a smoother energy landscape attributed to the restrained formation of energetically discrete low-$n$ phases. Moreover, the pristine film showed a gradual decrease in PLE intensity relative to the absorbance in the higher energy (shorter wavelength) region, while that of the A-IDP film was mostly maintained. This increased disparity between PLE and absorbance in the pristine film implies a considerable loss of excited charge carriers at energies well above the band gap,

contributing less to the light emission. Conversely, a relatively larger proportion of charge carriers in the A-IDP film contributes to light emission, suggesting improved funnelling of the excess photoexcited energy to the emitting state. As a result, the A-IDP films exhibited substantially increased PL intensity and PL lifetime compared to that of pristine films (Fig. 1g and Supplementary Fig. 7).

## Phase regulation by cation-π intermediate complex

The formation of the intermediate phase was triggered by incorporating highly reactive methylenediamonium dichloride (MDACl$_2$) into the precursor solution. In solution, MDACl$_2$ rapidly degrades into NH$_4^+$ and hexamethylenetetramine (HMTA)[20]. Exploiting this proposed reaction scheme, we induced the formation of an intermediate complex with PEA$^+$. First, the steady-state UV-Vis absorption spectra of mixture solution of MDACl$_2$ and PEA$^+$ revealed the emergence of a new

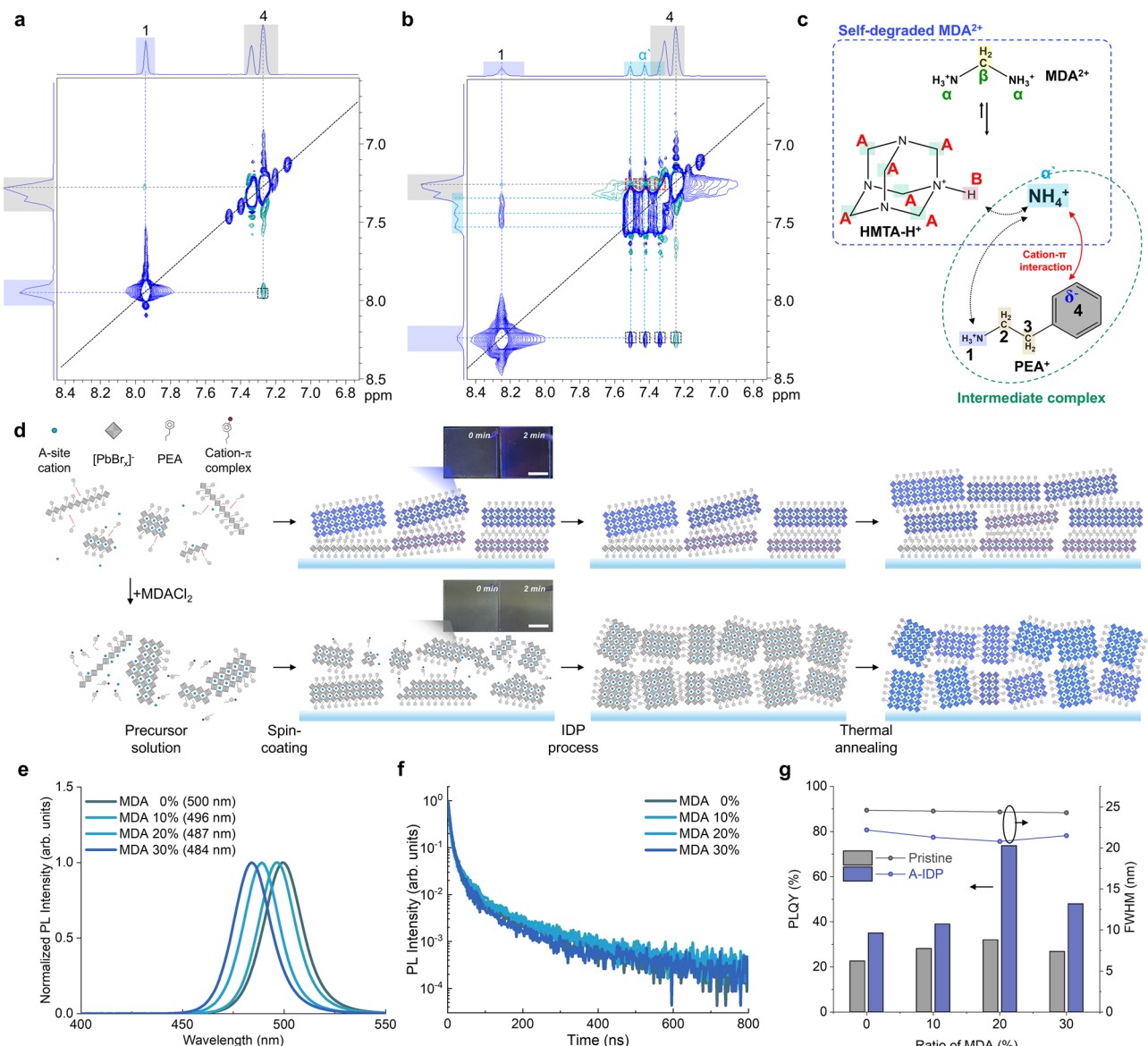

**Fig. 2 | Intermediate-assisted bright blue-emitting quasi-2D perovskites. a**, **b** NOESY 2D $^1$H NMR of perovskite precursor solution without MDACl$_2$ (**a**) and with MDACl$_2$ (**b**). **c** Schematic figure on the mechanism of cation-π intermediate complex formation. **d** Schematic illustration of crystallization process with (upper panel) and without (bottom panel) MDACl$_2$ in sequence of solution state, film state after spin-coating, IDP process, and thermal annealing process, each from left to right. Inset of the scheme after spin-coating: Picture of thin films with different time after spin-coating. Scale bar, 1 cm. **e** Steady-state PL spectra of quasi-2D perovskite thin films with different amount of MDACl$_2$ substitution. **f** Steady-state PL intensity and FWHM of perovskite films different amount of MDACl$_2$ substitution and A-IDP process. **g** PLQY and FWHM of perovskite films different amount of MDACl$_2$ substitution and with or without A-IDP process.

absorption band at ~400 nm as a result of formation of a new charge-transfer (CT) complex; considering no observation of CT complex among any of the precursor solutions or even with the pre-existing π-π stacking in PEABr-only solution, strong coordination with charge transfer is present in the MDA-PEA mixture solution with the additional electrostatic interaction with π electrons to form CT states[21] (Supplementary Fig. 8a–c).

To unveil the origin of the CT complex, we analysed the $^1$H nuclear magnetic resonance (NMR) spectra at different ratios of PEA$^+$ to MDACl$_2$ to obtain specific information on interactions among chemical species in solution (Supplementary Fig. 8d–i). Most peaks exhibited a slight downfield-shift (higher ppm) due to the increase in solution acidity with addition of MDA$^{2+}$. As the addition ratio of MDACl$_2$ increased, the peak associated with the -NH$_3^+$ group of PEA$^+$ shifted gradually downfield from 7.94 ppm to 8.24 ppm in same stoichiometric ratio; this result indicates deshielding of protons induced by the

acidified solution. The protons from NH$_4^+$ showed a 1:1:1 triplet peak (with splitting due to spin–spin coupling of $^1$H to the quadrupolar $^{14}$N nucleus), which also shifted downfield. In contrast, the C$_6$H$_5$ peak in PEA$^+$ shifted gradually upfield as the MDACl$_2$ ratio increased, despite the increasing acidity of the solution; this result suggests instead that additional shielding of protons occurred by coordination of π electrons with Brønsted–Lowry acid.

To determine the configuration of PEA$^+$ here, we conducted 2D $^1$H Nuclear Overhauser Effect Spectroscopy (NOESY), which shows through-space interactions of characteristic protons within 5 Å (Fig. 2a, b)[21]. The NOESY spectrum of pure PEA$^+$ solution showed a diagonal peak from self-association of each proton, and some cross-peaks that are related to interactions between adjacent protons of -NH$_3^+$ (7.94 ppm) toward the -CH$_2$- (2.8–3.0 ppm) moiety. In contrast, the spectrum of a PEA-MDA mixture solution showed additional cross-peaks between NH$_4^+$ (7.51–7.34 ppm) and C$_6$H$_5$ (7.214–7.323 ppm); these

peaks indicate the presence of cation-π interaction with spatially adjacent protons from two ions[22]. The binding energy of cation−π interaction between $NH_4^+$ and π electrons in $C_6H_5$ is reported to be significantly high, reaching up to ~18 kcal mol$^{-1}$, which can be arguably the strongest compared with any of the other noncovalent interactions in precursor solution, including hydrogen bonding, thereby coordinating $NH_4^+$ and $C_6H_5$[23,24]. The 2D Diffusion-Ordered Spectroscopy (DOSY) NMR also revealed that the diffusivity of $NH_4^+$ ions (~7.3 ppm) was decreased from $5.25 \times 10^{-10}$ m$^2$ s$^{-1}$ in the MDACl$_2$ solution to $4.85 \times 10^{-10}$ m$^2$ s$^{-1}$ in the PEABr/MDACl$_2$ mixture, indicating the coordination of $NH_4^+$ with π electrons in PEA$^+$ from the formation of a strong cation-π intermediate complex (Supplementary Fig. 9).

Combining these results, the coordination process can be visualized (Fig. 2c). The strong cation-π interaction of $NH_4^+$ with benzene effectively prevented the spontaneous crystallization of PEA$^+$ into low-$n$ phases (Fig. 2d). As expected, incorporation of MDACl$_2$ into precursor solution delayed the formation of low-$n$ phases and induced slow homogeneous growth after spin coating (Supplementary Fig. 10). Pristine film was already crystallized into photoactive quasi-2D phases without annealing process and thus showed blue emission under a UV lamp. In contrast, the MDACl$_2$-incorporated film showed negligible color and absorbance in the visible range; this change indicates an absence of any crystallized phase during this intermediate stage (**inset in** Fig. 2d).

The coordination chemistry between precursors forms metastable intermediate states, and thereby kinetically delay formation of the thermodynamically stable low-$n$ phases ($n \leq 3$). The films now have uniform distribution and maintain the perovskite crystals as an intermediate phase until the annealing process starts. Note that even with intermediate phases, the thermodynamic broadening of quasi-2D crystals toward low-$n$ phases can occur without the IDP process, as confirmed in GIWAXS result of pristine perovskite films with different amount of MDACl$_2$ (Supplementary Fig. 11).

The MDACl$_2$-included films with the A-IDP process showed distinct emission characteristics compared with pristine films. With a sufficient amount of MDACl$_2$ incorporated up to 20 mol. %, the uniform and narrowly distributed intermediate phase is achieved, which transformed into homogeneous medium-$n$ phases after A-IDP process. The PL spectra gradually blue-shifted from 500 nm toward 482 nm with narrowed emission spectra after A-IDP process (Fig. 2e). Importantly, the broad emission spectra with FWHM of >24 nm was observed with pristine films regardless of molar ratio of MDACl$_2$ (Supplementary Fig. 12). This indicates that merely facilitating formation of the kinetically stable intermediate complex is not enough to hinder the disproportion during thermal annealing process; a further requirement is that the complex must be pinned by excess solvent and ligand exchange process before thermodynamical broadening can occur to cause phase inhomogeneity and interfacial defects. Also, with the ideal pinning of medium-$n$ phases in a uniform kinetic stage, A-IDP perovskites with 20 mol. % MDACl$_2$ showed the longest PL lifetime, the highest PLQY of 73%, and a narrow emission spectrum with FWHM = 20.6 nm at 489 nm (Fig. 2f, g).

## Fast charge-funnelling in distribution-controlled phases

Full exploitation of the benefits of the A-IDP-engineered perovskite films requires understanding of the mechanism by which the intermediate-direct-pinned quasi-2D crystals affects the behavior of charge carriers. To investigate this phenomenon, we conducted time-resolved spectroscopic studies of the perovskite films. Charge-carrier funnelling in the perovskite films was directly assessed using femtosecond transient absorption (fs-TA) measurements. By pumping at $\lambda = 400$ nm, we generated a population of charge carriers at the energy state of the 2D perovskite crystals ($n = 1$). Pump fluence was kept low ($1.4 \times 10^{17}$ cm$^{-2}$) to minimize high-order recombination processes and avoid complicating the carrier-relaxation dynamics.

Immediately after photoexcitation, the initial ground state bleaching (GSB) signal within the $450 \leq \lambda \leq 490$ nm ($n = 3-5$) indicated that the charge carriers were rapidly funnelled to the lowest states of medium-$n$ phases (Fig. 3a–d). In the pristine film, the TA spectral evolution revealed the initial GSB maximum at 450 nm ($n = 3$) vanishing and evolving into 470 nm ($n = 4$) within hundreds of femtoseconds (Fig. 3c, e). This observation suggests state-to-state energy funnelling towards the emitting states at progressively lower energies. In stark contrast, in the A-IDP film, the GSB signal was initially absent at 450 nm ($n = 3$) and was mainly distributed near 488 nm ($n = 5$) (Fig. 3d, f); this skipping of the $n = 3$ state indicates highly-rapid funnelling of charge carriers to the lowest state, within the instrument's temporal resolution of 170 fs, which may have been triggered by different distribution of density of states (DOS) (Fig. 3g). Over hundreds of femtoseconds, the GSB signal increased and its peak gradually redshifted; these indicate the ongoing energy funnelling across the quasi-2D manifold.

The red edge of the GSB developed a derivative-like photo-induced absorption (PA) feature, which was pronounced in the A-IDP film but was negligible in the pristine film (Fig. 3c, d and Supplementary Fig. 13a, b). One origin of this PA feature may be the relatively reduced sub-bandgap defect states in A-IDP film compared to that in pristine film, as indicated by the lower absorbance of the red tail in UV-Vis spectra (Fig. 1f). The weaker GSB signal on the lower energy side of excitonic transitions may have contributed to the more prominent appearance of this PA feature. Another possibility is that the PA feature arises from the attractive exciton-exciton interaction[25], and the observed stronger exciton-exciton interaction in A-IDP film evidences for a relatively random orientation of the crystals within the film. In quasi-2D perovskites, the organic layers tend to dielectrically confine the exciton within the inorganic crystal along the out-of-plane direction. Excitons that are strongly confined within the crystal are well isolated dielectrically, reducing the likelihood of interaction with the excitons in neighboring crystals, especially when crystals are orderly packed. However, when the crystals are randomly oriented, the dielectric confinement experienced by an exciton is partially canceled out by the differently oriented neighboring crystals. This reduction in dielectric screening increases the likelihood of interaction between excitons in adjacent crystals. These overall improve the spatial accessibility of excitons in neighboring inorganic octahedral layers, greatly facilitating the energy funnelling across the excitonic manifold in A-IDP film and ultimately benefiting the LED device efficiency.

To elucidate the specific pathway and rate of charge carrier funnelling, TA kinetics were analysed at each energy states of $\lambda = 450$ nm, 470 nm, and 488 nm corresponding to the $n = 3$, 4, and 5, respectively (Fig. 3e, f). In the pristine film, the charge carriers initially accumulated at $n = 3$ and rapidly decayed, following a double-exponential function with decay-time constants of 220 fs and 400 fs (Fig. 3e). These decay constants perfectly matched with the rise time constants of the $n = 4$ and $n = 5$ state populations, directly verifying the carrier-funnelling pathway from $n = 3$ to $n = 4$ (220 fs) and from $n = 3$ to $n = 5$ (400 fs). In contrast, the $n = 4$ population showed negligible decay with a slow time constant of 1,500 fs; this slow rate indicates inefficient carrier funnelling from $n = 4$ to $n = 5$ in the pristine quasi-2D perovskite despite their energetic proximity. This inefficiency is attributed to the lower proportion of $n = 5$ states and the unfavorable (well-ordered, dielectrically screened) orientation of quasi-2D crystals in the film.

In contrast, the A-IDP film demonstrated distinct behavior, with most of the carrier population initially accumulating at $n = 5$ with a rise-time constant of 264 fs, which was 1.5 times faster compared to the pristine film's 400-fs funnelling time to $n = 5$ (Fig. 3e, f). Furthermore, charge carriers accumulated at $n = 4$ showed a 400-fs decay, concurrent with the second rise-constant of the $n = 5$ signal, effectively depleting the carrier population at $n = 4$ by efficient energy funnelling. The carrier population was negligible at $n = 3$; because of this, the $n = 3$

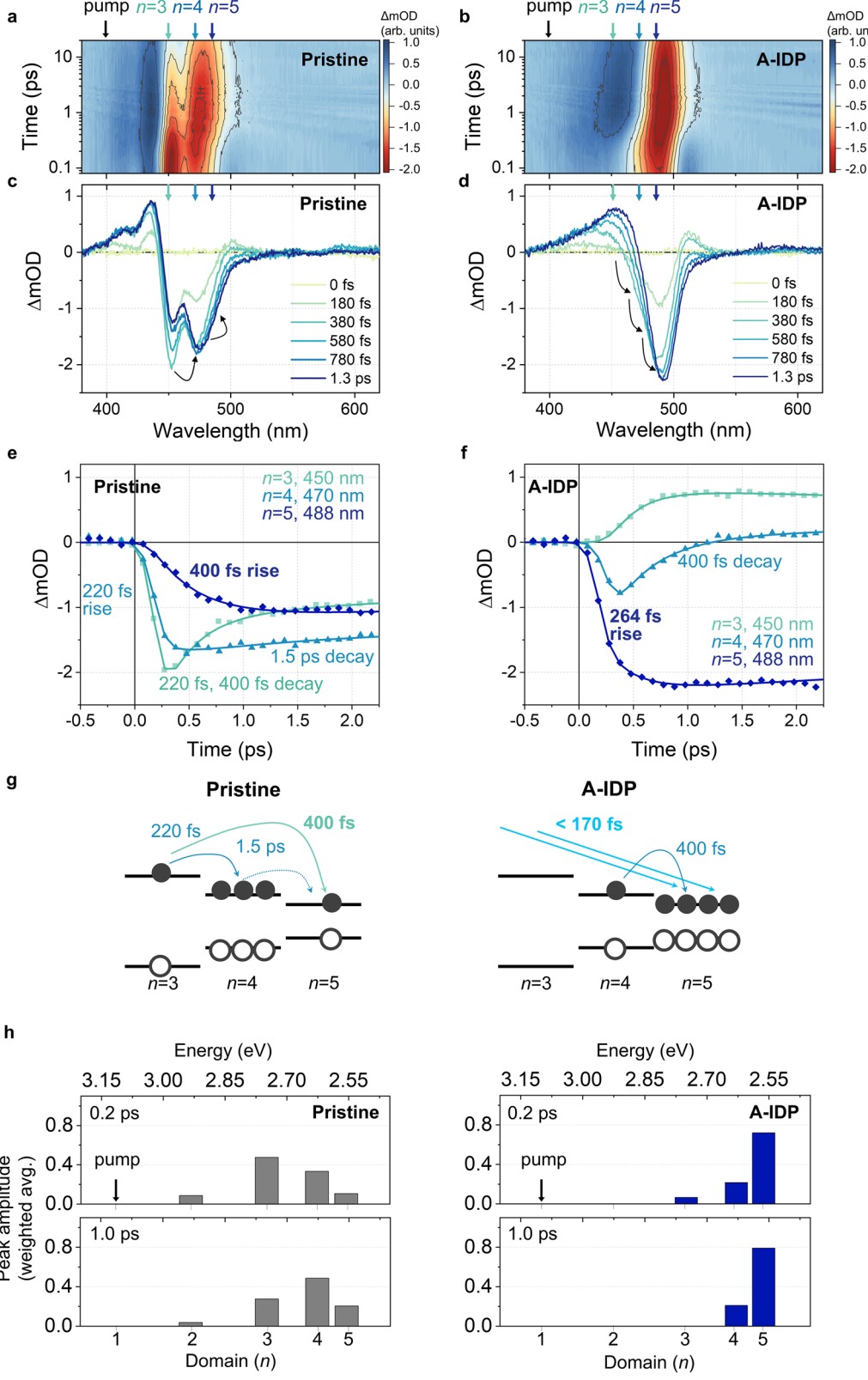

**Fig. 3 | Femtosecond TA spectra of pristine and A-IDP perovskite films. a, b** 2D contour plot and **c, d** early-time spectra of pristine (**a, c**) and A-IDP (**b, d**) films. **e, f** TA kinetic traces of pristine (**e**) and A-IDP (**f**) perovskite films probed at 450 nm, 470 nm, 488 nm, each corresponding to domains of $n = 3,4,5$, respectively. **g** Schematic description of the charge carrier funnelling occurring in pristine and A-IDP films. The DOS were qualitatively estimated from the initial carrier population distribution (TA spectra at $t = 0.2$ ps). **h** Carrier population distribution of pristine and A-IDP films presented by the weighted average peak area of TA spectra at delay times 0.2 ps (upper panels) and 1 ps (lower panels).

kinetic trace was overwhelmed by the excited-state absorption (ESA) signal, the dynamics of which merely followed the decay of the $n = 4$ state. Thus, the charge-carrier funnelling in the A-IDP film differed from that in the pristine film by (i) direct funnelling of carriers mostly to $n = 5$ with no accumulation at $n = 3$ (within the temporal resolution) and (ii) effective funnelling away of the remaining carriers at $n = 4$ (Fig. 3g).

To further clarify the energy-funnelling pathway, we analysed the charge-carrier distribution across different $n$ domains over time using the reported quantification method[10]. We fitted the TA spectra (Fig. 3c, d) using multiple Gaussian functions to quantify the contribution of each domain's transition (Supplementary Fig. 14). The population distribution was initially obtained at a delay time of 0.2 ps (upper panel of Fig. 3h). In the pristine film, charge carriers predominantly accumulated at $n = 3$, whereas in the A-IDP film, most of the charge carriers were efficiently funnelled to the $n = 5$ state. At a delay time of 1 ps, when the dynamics of hundreds of femtoseconds had been nearly completed, the carrier population at $n = 3$ in the pristine film was partly shifted to $n = 4$ and $n = 5$, but the population remained smaller at $n = 5$ than at $n = 4$ ($n = 5$ to $n = 4$ population <1). In contrast, in the A-IDP film, at a time delay of 1 ps, the $n = 3$ population was fully depleted by funnelling, and the population ratio of $n = 5$ to $n = 4$ had increased to nearly 4:1.

Additional fs-TA measurements were conducted to investigate the function of the CCA as an anchor molecule at the crystal surface. TA dynamics of the S-IDP film without the CCA molecule was compared with that of the A-IDP film with the CCA molecule. The difference in TA dynamics between two films was pronounced at higher pump fluence ($4.1 \times 10^{17} \, cm^{-2}$). The initial TA spectrum of the S-IDP film showed a GSB shoulder at 450 nm, which was absent in the A-IDP film; this evidences a partial accumulation of charge carriers at $n = 3$ in the S-IDP film (Supplementary Fig. 15a, b). Furthermore, introduction of CCA clearly shortened the decay constant of the $n = 4$ state in A-IDP (370 fs) compared to that in S-IDP (434 fs) (Supplementary Fig. 15c) and also the corresponding rise constant of $n = 5$ state in A-IDP (354 fs) compared to that in S-IDP (456 fs), showing the accelerated carrier funnelling towards $n = 5$ (Supplementary Fig. 15d).

The carrier population distributions of A-IDP and S-IDP films clearly illustrate the energy funnelling pathway, as seen in both films at 0.2 ps, where the initial distribution was similar, besides a smaller $n = 3$ contribution in A-IDP film than in S-IDP film (Supplementary Figs. 15e and 13). However, at 1 ps, a larger portion of the $n = 4$ population had been funnelled into $n = 5$ in the A-IDP film compared to that in the S-IDP film. Such a difference indicates that the presence of the CCA additive molecule increases the overall energy-funnelling rate by specifically accelerating the $n = 4$ to $n = 5$ pathway. CCA substitutes PEA and binds to perovskite layers with a smaller molecule size, thereby reducing the Van der Waals gap between neighboring quasi-2D perovskite crystals. This additive effect was more clearly observed as the pump fluence was increased, which suggests that the effect of CCA was more pronounced in the presence of the phonon-bottleneck effect. Smaller Van der Waals gap between the quasi-2D crystals reduces the density of states for phonons, and thereby facilitates the charge-carrier funnelling even at high initial carrier density[26,27].

Therefore, we suggest that the rearrangement of crystals upon IDP treatment, whether with or without the additive, smooths the energetic landscape of the film and facilitates carrier funnelling into domains with higher $n$ by preventing charge accumulation in lower $n$ domains. Further, the inclusion of CCA molecules chemically optimized the crystal orientations within the film to further facilitate interdomain charge transfer, which was found to be particularly effective between the medium-$n$ phases ($n = 4, 5$). Thus, the efficient energy funnelling in optimized IDP quasi-2D perovskite films in this study is attributed to the interplay between the engineered energy landscape, the reorientation of quasi-2D crystals, and the finely-tuned Van der Waals gap between adjacent quasi-2D crystals.

## Efficient and narrow sky-blue emission

To exploit the dominant phase distribution of medium-$n$ phases and the fast charge-transport characteristics, we developed sky-blue emitting PeLEDs that use A-IDP quasi-2D perovskites (Fig. 4a). The development of sky-blue LEDs represents an important step toward achieving efficient deep-blue emission. This approach mirrors the strategy used in the development of stable blue organic light-emitting diodes (OLEDs) with a microcavity structure, where sky-blue OLEDs were preferred due to their much longer lifetimes[28,29]. Similarly, given the short lifetimes typically observed in deep-blue PeLEDs, sky-blue PeLEDs−offering longer lifetimes and higher efficiency−could be advantageous until stable and efficient deep-blue PeLEDs are realized. These devices, when combined with a microcavity effect that shifts the sky-blue spectra towards deeper blue, present a promising approach. From this perspective, developing an ideal material system for sky-blue emission is critical, as it can enable efficient and stable deep-blue emission following optical cavity engineering. Moreover, stable and narrow sky-blue light sources are valuable for their role in creating efficient and stable white LEDs when combined with yellow phosphors. Specifically, narrow backlight emission provides two key advantages: (1) higher power efficiency and (2) better spectral stability of white emission by avoiding unintended backlight leakage in wavelength ranges that the phosphor cannot absorb[30].

Transient EL measurement of PeLEDs showed faster rising time at the onset of the voltage pulse and EL decay time after the voltage pulse off in A-IDP PeLEDs than in S-IDP and pristine PeLEDs, as a result of accelerated carrier transfer in A-IDP PeLEDs (Supplementary Fig. 16a−c). The A-IDP perovskites films also exhibited an order higher lateral conductivity ($\sigma$) compared to that of pristine perovskite film (Supplementary Fig. 17). Benefiting from the high PLQY and accelerated charge transport, the A-IDP PeLEDs showed maximum EQE of 22.5% (current efficiency = 26.62 cd A$^{-1}$) at $\lambda = 489$ nm (FWHM = 20.8 nm), calculated using the full angular EL distribution (Fig. 4b−d, Supplementary Fig. 18 and Supplementary Table 1). The summarized EL characteristics of pristine and A-IDP PeLEDs with different amounts of MDACl$_2$ and CCA agree well with the tendency in PL characteristics (Supplementary Figs. 19, 20 and Supplementary Table 2). We further increased the EQE of our PeLEDs to 41.8% by placing a light-outcoupling hemispherical lens on the emitting glass substrate. Compared to pristine PeLEDs, which exhibit a turn-on voltage greater than 3.0 V, both S-IDP and A-IDP PeLEDs showed a sub-gap turn-on voltage of 2.2 V−below the bandgap ($E_g = 2.54$ eV). This observation can be attributed to the recombination of non-thermal-equilibrium band-edge carriers in the emitting layer, where the carrier populations are perturbed by a small external bias, leading to radiative recombination below the turn-on voltage[31,32]. Furthermore, PeLEDs processed via the IDP method benefited from improved charge transport and recombination of band-edge carriers, which can be attributed to the suppression of charge accumulation in low-$n$ phases. Also, the EQE histogram of A-IDP PeLEDs (average = $19.7 \pm 1.2$%, 37 devices) shows that our strategy can provide highly reproducible quasi-2D perovskite thin films for blue emission (Fig. 4e).

These PeLEDs are an advance in that previously-reported high-efficiency PeLEDs have been largely limited to green emission by using single-halide composites. Previous PeLEDs that used mixed-halide composite, which is important to induce a blue shift, showed significantly decreased emission efficiency at $\lambda < 500$ nm due to halide segregation and the resulting inhomogeneity in the energy landscape. In contrast, by homogeneously anchoring medium-$n$ quasi-2D phases, we achieved a remarkably high EQE of 22.5% in the sky-blue emission region (Supplementary Table 3). Moreover, remarkably narrow EL

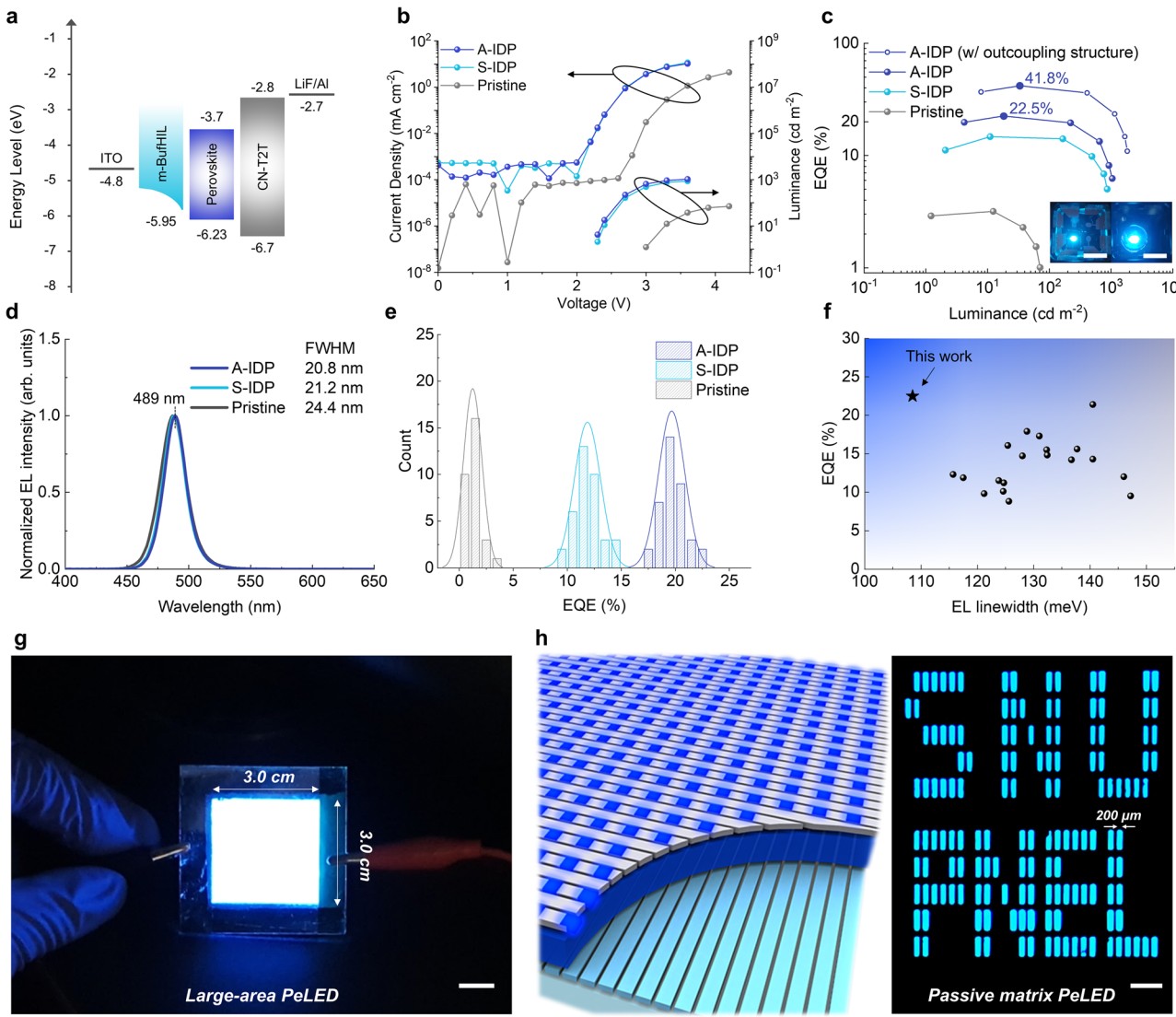

**Fig. 4 | EL characteristics of blue PeLEDs. a** schematic energy diagram of PeLEDs. **b** luminance and current density vs voltage, **c** EQE vs current density (inset: operating PeLEDs without (left) and with (right) hemispherical lens, scale bar: 1 cm), **d** normalized EL spectra, **e** EQE histogram of PeLEDs. **f** Characteristics of reported blue PeLEDs with peak emission wavelength <495 nm based on maximum EQE and EL linewidth. Summarized in Supplementary Table 3. **g** Photograph of operating large-area PeLEDs (pixel size: 900 mm²). Scale bar: 1 cm. **h** Schematic structure and picture of operating 30 × 10 passive-matrix PeLEDs (active area = 200 μm × 600 μm). Scale bar: 1 mm.

linewidth (108 meV) of our PeLEDs among efficient quasi-2D blue PeLEDs (EQE > 15%) evidences the advantage of ideal energy landscape in quasi-2D emitter structure achieved by the A-IDP method (Fig. 4f). The A-IDP PeLEDs also showed stable EL operation without significant spectral shift ($\Delta\lambda$ ~ 1 nm at 1000 cd m$^{-2}$), compared with that of pristine PeLEDs ($\Delta\lambda$ ~ 2 nm at <100 cd m$^{-2}$) (Supplementary Fig. 16d). We also fabricated PeLEDs with emission spectra from deep-blue (469 nm) to green (504 nm) region based on A-IDP process; these have EQEs of 5.7% for 469 nm and 12.5% for 479 nm, and 21.7% for 504 nm, respectively (Supplementary Fig. 21).

Device stability of the PeLEDs were compared by monitoring the luminance under constant current operation. Compared with the pristine PeLEDs having half-lifetimes of $T_{50}$ = 1 min at initial brightness $L_0$ = 50 cd m$^{-2}$, A-IDP PeLEDs showed 150 times longer $T_{50}$ = 2.5 h at $L_0$ = 100 cd m$^{-2}$, realizing remarkably long operational half-lifetime among PeLEDs with $\lambda$ < 500 nm (Supplementary Fig. 16e, f). Benefiting from the enhanced stability and ultralow turn-on voltage, we fabricated large-area blue PeLEDs (pixel size: 900 mm²) successfully

with high uniformity (Fig. 4g). Also, we developed the first blue-emitting passive-matrix PeLEDs with 30 × 10 arrays (active area = 200 μm × 600 μm) by photolithographic ITO electrode patterning, providing uniform blue emission among adjacent pixels. (Fig. 4h). These highlights the potential applications of PeLEDs to addressable full-color panel displays.

We demonstrated an anchoring-assisted intermediate direct pinning (A-IDP) process to achieve efficient and narrow-emitting quasi-2D perovskite structure. The A-IDP process enabled formation of medium-$n$ phases ($n$ = 4–5) by removing excess large cation and by kinetically pinning the uniformly-distributed intermediate phases. The A-IDP process uses a CCA molecule, which became incorporated as a molecular anchor at the surface of quasi-2D crystals, and maximized the charge transport between crystals. The perovskites produced using A-IDP showed improved luminescent efficiency and fast charge funnelling toward medium-$n$ phases. As a result, we developed efficient blue PeLEDs at 489 nm with maximum EQE = 22.5%, the narrow EL linewidth of 108 meV among quasi-2D blue PeLEDs, and greatly

improved device operational lifetime. We extended our approach to the first demonstration of blue-emitting passive matrix LEDs ($30 \times 10$ arrays, active area $= 200\,\mu m \times 600\,\mu m$) with great pixel clarity. Our blue PeLEDs demonstrate the great potential of high-efficiency perovskite emitters for narrow blue emission and provide valuable insights to guide further development of blue emission, which can be exploited to realize full-color displays.

## Methods

### Materials

Phenethylammonium bromide (PEABr) and ethylammonium bromide (EABr) were purchased from Greatcell Solar. Cesium bromide (CsBr), methylenediamine dihydrochloride ($MDACl_2$), γ-Aminobutyric Acid (GABA), Poly (sodium-4-styrenesulfonate) (PSSNa), dimethyl sulfoxide (DMSO), anisole, and tetrahydrofuran (THF) were purchased from Sigma-Aldrich. Lead bromide ($PbBr_2$) and 1,2-cis-cyclohexane dicarboxylic acid (CCA) were purchased from Tokyo Chemical Industry (TCI) Co. Ltd. 3′,3′′′,3′′′′′-(1,3,5-triazine-2,4,6-triyl) tris(([1,1′-biphenyl]-3-carbonitrile)) (CN-T2T) for the ETL was purchased from Lumtec. Lithium fluoride (LiF) was purchased from FOOSUNG Co., Ltd.

### Preparation of solution

PEA-CS-EA-$MDACl_2$ mixed precursor was made by dissolving PEABr, CsBr, (EABr$_{1-x}$: ($MDACl_2$)$_x$), and $PbBr_2$ in a molar ratio of 2: 1.6: 2.4: 3 in DMSO; the final concentration was 0.25 M. EABr is substituted with $MDACl_2$. CCA was dissolved in THF at a concentration of $2\,g\,L^{-1}$.

### Perovskite light-emitting diodes fabrication

Glass substrates with a 70 nm indium tin oxide (ITO) layer underwent sequential ultrasonic cleaning in acetone and isopropyl alcohol (IPA) for 15 min each, followed by a 15-min boiling in IPA. Subsequently, the substrates were exposed to ultraviolet ozone for 15 min. Then 30 nm of a modified buffer hole injection layer (m-Buf-HIL) with a gradient work function was deposited by spin-coating mixture of PEDOT: PSS (CLEVIOS P VP AI4083) and PFI copolymer in a 1:1.5 weight ratio with additional PSSNa ($2\,g\,L^{-1}$) and GABA ($2\,g\,L^{-1}$) as additive molecules for better coordination with the perovskite precursors during crystallization[33,34]. The resulting films were thermally treated at 150 °C for 30 min to complete the annealing process. The annealed samples were subsequently moved into a nitrogen-filled glove box, where the emissive layers were deposited via spin-coating. Quasi-2D precursor solutions were spin-coated at 7000 rpm for 60 s with anti-solvent engineering using anisole. For the A-IDP process, the CCA solution was poured onto the intermediate film, immediately spin-dried, and then annealed at 80 °C for 2 min. The samples were then transferred to the thermal evaporator under high vacuum ($< 10^{-7}$ torr) and sequentially deposited CN-T2T (40 nm), LiF (1 nm), and Al (100 nm) on top of the emissive layer.

### Characterization of perovskite films

Nanoscale images of the perovskite thin films were obtained using a field-emission scanning electron microscope (SEM) (SUPRA 55VP). For the XPS and UPS spectra, a photoelectron spectrometer (AXIS-Ultra DLD, Kratos Inc.) was used with a monochromatic Al-Kα line (1486.6 eV) for XPS and He I radiation (21.2 eV) for UPS analysis. Steady-state PL spectra of the perovskite thin films were measured using a spectrofluorometer (JASCO FP-8500) with encapsulation under nitrogen atmosphere. The excitation wavelength was 330 nm (Xenon arc lamp; power = 150 W). For the transient PL decay measurements, a streak camera (c10627, Hamamatsu) and a nitrogen pulse laser (337 nm, 20 Hz, Usho) was combined to obtain the transient and spectral emission behavior of the samples being monitored at the same time.

### Transient absorption spectroscopy measurement

The fs-TA spectrometer used in this study consists of an optical parametric amplifier (OPA; Palitra, Quantronix), which is pumped by a Ti: sapphire regenerative amplifier system (Integra-C, Quantronix) with a repetition rate of 1 kHz and an optical detection part. The OPA generated pulses with a width of 100 fs within the 280–2700 nm wavelength range, serving as the pump pulses. White-light continuum (WLC) probe pulses were generated using a 3-mm-thick $CaF_2$ window by focusing a small portion of the fundamental 800 nm pulses. The time delay between these pump and probe pulses was precisely regulated by adjusting the probe beam path using a variable optical delay (ILS250, Newport). The spectrally dispersed WLC probe pulses were then captured using a high-speed spectrometer (Ultrafast Systems). To acquire the time-resolved transient-absorption difference signal ($\Delta A$) at a specific delay time, the pump pulses were chopped at 500 Hz, to alternately record the absorption spectra intensities with or without the pump pulses. The polarization angle between pump and probe beams was set at the magic angle (54.7°) using a Glan-laser polarizer and half-wave retarder to disregard the polarization-dependent signals. Measurements were performed by shaking both the samples and the $CaF_2$ window to minimize the photobleaching/photodegradation effect.

### Grazing-incidence X-ray diffraction measurements

GIWAXS was performed on the perovskite thin films at beamlines 6D and 9 A of the Pohang Accelerator Laboratory (PAL), Republic of Korea. The incident angle of the X-ray beam ($\lambda = .0722$ Å) on a sample was fixed at 0.12°.

### Device characterization

Current density-voltage-luminance ($J$-$V$-$L$) characteristics of the PeLEDs were measured using a Keithley 236 source measure unit and a Minolta CS 2000 spectroradiometer. The PeLEDs were measured from zero bias to forward bias under a scanning rate of $0.05\,V\,s^{-1}$. The EQE of PeLEDs were calculated by measuring their full-angular electroluminescence distributions[35]. Transient EL characteristics of PeLEDs were monitored by using a streak camera system composed of a streak scope 326 (C10627, Hamamtsu Photonics), a CCD camera (C9300, Hamamatsu Photonics), a delay generator 327 (DG645, Stanford Research Systems), and a function generator (33220 A, Agilent). Operational lifetime of PeLEDs was measured under constant current condition by simultaneously tracking brightness and applied voltage by using a M760 Lifetime Analyzer (McScience Inc.) with a control computer under air-conditioned environment <18 °C.

## Data availability

The data that support the findings of this study are available from the corresponding authors upon request.

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

## Acknowledgments

This work was supported by the National Research Foundation of Korea (NRF) grant funded by the Ministry of Science and ICT (MSIT) (RS202500560490, 2022M3H4A1A04096380) and NRF grant Brain Link program (2022H1D3A3A01081288). This work was also supported by the NRF through a grant funded by the South Korean government (MEST) (No. 2021R1A2C300630813) and Korea Technology and Information Promotion Agency (TIPA) grant funded by Korea government Ministry of SMEs and Startups (MSS) (No. S3396074).

## Author contributions

J.S.K., H.-D.L., J.H.J., and T.K. equally contributed to this work. H.-D.L., J.H.J., and T.-W.L. initiated the study and J.S.K., H.-D.L., J.H.J., and T.-W.L. designed the study. J.S.K., H.-D.L., J.H.J., and S.E.C. fabricated LED devices and analysed data. H.Y. conducted GIWAXS measurement and data analysis. T.K. and D.K. designed and conducted the spectroscopic experiments, including UV-Vis absorption, PLE, and femtosecond transient absorption spectroscopy and analysed and interpreted the excited state dynamics. H.J.Y. conducted the XPS analysis. J.S.K. conducted the NMR data analysis. J.M. and C.-m.K. conducted integration and analysis of passive-matrix PeLEDs. D.-H.K. conducted the FTIR analysis. S.-J.W. conducted the transient PL analysis. S.K. conducted the device analysis using the light outcoupling structure. T.-W.L. supervised the work. J.S.K. drafted the first version of the manuscript with assistance from T.-W.L. All authors discussed the results and commented on the manuscript.

## Competing interests

The authors declare no competing interests.
