## [Transparent Peer Review file · Nature Communications]

Kinetically-controlled intermediate-direct-pinning for homogeneous energy landscapes in quasi-two-dimensional perovskites for efficient and narrow blue emission

Corresponding Author: Professor Tae-Woo Lee

Version 1:

Reviewer comments:

Reviewer #2

(Remarks to the Author)

The authors have revised the manuscript accordingly, and the paper is ready for publication now.

Reviewer #3

(Remarks to the Author)

In this manuscript, Kim et al. reported an intermediate-direct-pinning strategy to fabricate the quasi-2D perovskite with homogeneous phase distribution. By introducing the CCA molecules to retard the rapid crystallizations of low n-value phases, they achieved sky-blue-emitting films with narrowest emission linewidth (108 meV) and efficient PeLEDs with maximum external quantum efficiency of 22.5%. The formation mechanisms of intermediate phase and the related crystallization kinetics were both well illustrated, and the performance of the resulting PeLEDs was impress. Thus, I recommend to publish the manuscript in Nature Communications after the following issues have been well addressed. The detailed comments are as followed:

- (1) How about the universality of this strategy? Whether the n-value phases can be controllably tuned, such as $n = 2, 3$ to achieve deep-blue emission?
- (2) The authors should highlight the advancement of this strategy. The linewidth was 108 meV (i.e. FWHM was around 21 nm), which was similar with previous reports (Nature 2021, 599, 594; Angew. Chem. Int. Ed. 2023, 62, e202302184; Adv. Sci. 2022, 9, 2201807; Sci. Adv. 2024, 10, eado5645).
- (3) Whether the CCA treatment can influence the orientation of quasi-2D perovskite phases? As shown in Extended Data Fig. 2b-d, the longer the CCA-THF solution exposure time was, the sharper Bragg spots in the diffraction rings were. Additionally, is the diffraction ring at around 0.9 \AA^{-1} assigned to quasi-2D phases in Fig. 1c? Why dose this diffraction peak diminish as prolonging CCA-THF solution exposure time (Extended Data Fig. 2b-d) but reappear after annealing process (Fig. 1c)?
- (4) The authors mentioned that in pristine and A-IDP films, the ground state bleaching (GSB) peaks of different n-value phases were probed at same locations (Fig. 3e-f). It seems contradictory that the CCA can intercalate into the lattice of quasi-2D perovskites (Fig. 1). As well documented, the mixed spacer cations would change the bandgap of quantum wells (Nat. Commun. 2018, 9, 3541; Adv. Mater. 2021, 2104381), leading to the slight shifting of GSB peaks.
- (5) The turn-on voltage of S-IDP and A-IDP PeLEDs were both achieved at 2.2 V, which were quite lower than the bandgap ($E_g = \sim 2.54 \text{ eV}$). The corresponding device simulations and sub-bandgap turn-on mechanism should be discussed.
- (6) The related performances and efficiency calculations of PeLEDs by placing on the light-outcoupling hemispherical lens should be provided. Whether the emission still follow Lambert's cosine law?

Version 2:

Reviewer comments:

Reviewer #3

(Remarks to the Author)

I recommend to publish it on Nature Communications, and I have no further comments.

Responses to Reviewers

First of all, we would like to thank the reviewers for their valuable comments and questions. We have revised our manuscript in response to these comments with additional structural, optical and device characterizations. We could also verify that our approach can provide the most homogeneous energy landscape among quasi-2D PeLEDs in terms of emission linewidth, providing high EQE and long operational lifetime at the same time. Also, we provided that our additive-induced intermediate direct pinning (A-IDP) approach for PeLEDs can be universally applied to emission spectra of green to deep-blue emission range, achieving maximum EQE of 5.7% at deep-blue emission (<470 nm), EQE of 12.5% at pure-blue emission (<480 nm), EQE of 22.5% at sky-blue emission (489 nm), and EQE of 21.7% at green emission (504 nm) with A-IDP method. Lastly, we thoughtfully reviewed and confirmed the device efficiency calculation process with incorporation of hemisphere lens for maximizing the outcoupling efficiency of our PeLEDs, by cross-checking the outcoupling efficiency with both integrating sphere and spectroradiometer setup. We marked the revised parts of our manuscript in red.

Reviewer #2 (Remarks to the Author):

The authors have revised the manuscript accordingly, and the paper is ready for publication now.

Response) We are pleased that our revisions have fully addressed the previous comments which greatly improved the manuscript. We sincerely thank the reviewer again for the positive and constructive comments.

Reviewer #3 (Comments for the Author):

In this manuscript, Kim et al. reported an intermediate-direct-pinning strategy to fabricate the quasi-2D perovskite with homogeneous phase distribution. By introducing the CCA molecules to retard the rapid crystallizations of low n-value phases, they achieved sky-blue-emitting films with narrowest emission linewidth (108 meV) and efficient PeLEDs with maximum external quantum efficiency of 22.5%. The formation mechanisms of intermediate phase and the related crystallization kinetics were both well illustrated, and the performance of the resulting PeLEDs was impress. Thus, I recommend to publish the manuscript in Nature Communications after the following issues have been well addressed.

Response) First, we sincerely thank the reviewer for the detailed comments. To comply with them, we thoroughly revised the manuscript after supplementing the discussions regarding optical and structural characterization, device physics, and universality. Please see below for our point-by-point responses.

Q1. How about the universality of this strategy? Whether the n -value phases can be controllably tuned, such as $n = 2, 3$ to achieve deep-blue emission?

Response) We sincerely thank the reviewer for these constructive comments. As noted, adjusting the n -value phases to have $n = 2, 3$ can be beneficial for achieving a larger bandgap suitable for deep-blue emission. However, the focus and benefit of our research is developing both 1) homogeneous energy landscape and 2) fast charge transfer by exploiting the medium- n phases ($n = 4, 5$): both the characteristics cannot be achieved by tuning the n -value toward low- n phases ($n \leq 3$) with large amount of insulating ligand moieties. Instead, we could provide that our approach on medium- n phases also has universal applicability for bandgap tuning toward the deep-blue region by varying the halide ratio. Rather than focusing on low- n phases, we demonstrated PeLEDs based on medium- n phases with emission spectra ranging from green to deep blue by different Bromide (Br) to Chloride (Cl) halide ratio (**Extended Data Fig. 9**). With the controlled ratio of Cl, we could fabricate PeLEDs with a maximum EQE of 5.7% at a peak wavelength of $\lambda = 469$ nm (Commission Internationale de L'Eclairage (CIE) coordinates of (0.125, 0.071)), 12.5% at $\lambda = 479$ nm (CIE coordinates of (0.106, 0.122)), and 22.5% at sky-blue emission (489 nm). Additionally, by incorporating naphthalene sulfonic acid (NSA)—a stronger anchoring molecule—we realized further high- n domains for a smaller bandgap, thereby fabricating efficient green PeLEDs with an EQE of 21.7% at $\lambda = 504$ nm. Leveraging this material engineering strategy, our approach can be extended to other emission wavelengths while maintaining the homogeneous energy landscape. To provide the universality of our approach on wavelength tunability, we added the device J-V-L characteristics of PeLEDs over deep-blue to green emission region in the manuscript as below.

Revised parts in the manuscript

In the page 14, line 21

“We also fabricated PeLEDs with emission spectra from deep-blue (469 nm) to green (504 nm) region based on A-IDP process; these have EQEs of 5.7% for 469 nm and 12.5% for 479 nm, and 21.7% for 504 nm, respectively (**Extended Data Fig. 9**).”

Extended Data Fig. 9 | Emission wavelength tunability of A-IDP PeLEDs. **a**, Current density versus voltage, **b**, luminance versus voltage, **c**, EQE versus luminance, **d**, normalized EL spectra of A-IDP PeLEDs with emission peak center at 504 nm. **e**, The CIE coordinates of A-IDP PeLEDs with emission spectra from green (504 nm) to deep-blue (469 nm) region. The green PeLEDs were realized by substituting CCA with naphthalene sulfonic acid (NSA) to facilitate formation of further high-*n* phases with smaller band gap, while the deep-blue and blue emitting PeLEDs were achieved by increased molar proportion of chlorides.

Q2. The authors should highlight the advancement of this strategy. The linewidth was 108 meV (i.e. FWHM was around 21 nm), which was similar with previous reports (Nature 2021, 599, 594; Angew. Chem. Int. Ed. 2023, 62, e202302184; Adv. Sci. 2022, 9, 2201807; Sci. Adv. 2024, 10, eado5645).

Response) Thank you for your thoughtful comments and for highlighting the key advancement of our work. We agree with the reviewer's comment that the full-width at half maximum (FWHM) of emission peaks in this work can be similar across previous reports on blue PeLEDs; however, we'd like to highlight that the best way to compare the homogeneity of the energy landscape is the EL linewidth (meV) rather than the FWHM (nm) since the FWHM should decrease at larger bandgap even with same emission linewidth, which is the direct parameter providing distribution of energy states that contributing in the light emission (**Fig. R1a**). Hence, this refined metric of EL linewidth is strongly correlated with the actual distribution of energy landscape for achieving high EQEs and long operational lifetimes.

In this context, our work provides the high EQE and longest lifetime with the narrowest linewidth (108 meV) among efficient blue PeLEDs based on quasi-2D structure (**Fig. R1a-c**). The significantly higher EQE and half-lifetime of our work presents significant advancements in achieving efficient and stable blue emission in reduced-dimensional mixed-halide perovskites, addressing two critical challenges that have hindered progress in this field: dimensional inhomogeneity leading to phase broadness and halide segregation exacerbating the broadening of the energy landscape. Although recent reports on blue PeLEDs based on mixed-halide nanograin structure or QDs provided high EQE and similar level of narrow EL linewidth at the same time, their instability still lack far behind our approach, indicating the advantage of our strategy on great charge transport and recombination process from homogeneous energy landscape.

a

b

c

Fig. R2 | Trend of sky-blue to deep blue quasi-2D PeLEDs.

Summary of the reported sky-blue to deep blue quasi-2D PeLEDs (peak emission wavelength < 495 nm) characteristics based on **a**, 2D contour plot of EL linewidth based on peak wavelength versus FWHM, **b**, maximum EQE versus EL linewidth and **c**, half-lifetime versus EL linewidth.

Q3. Whether the CCA treatment can influence the orientation of quasi-2D perovskite phases? As shown in Extended Data Fig. 2b-d, the longer the CCA-THF solution exposure time was, the sharper Bragg spots in the diffraction rings were. Additionally, is the diffraction ring at around 0.9 \AA^{-1} assigned to quasi-2D phases in Fig. 1c? Why does this diffraction peak diminish as prolonging CCA-THF solution exposure time (Extended Data Fig. 2b-d) but reappear after annealing process (Fig. 1c)?

Response) We appreciate the reviewer for pointing this out and detailed comments on the structural aspects. To discuss the comments on Grazing incidence wide-angle X-ray scattering (GIWAXS) patterns accurately, we'd like to point out that every GIWAXS features of perovskite films were measured after the whole fabrication procedure including the annealing process (**Fig. R2**). Here in the previous version of manuscript, **Fig. 1b,c** is identical with **Extended Data Fig. 2a,b**, and the diffraction ring at 0.9 \AA^{-1} is not reappearing after annealing process, it's only diminishing with prolonged D_E from D_E of 0s (**Fig. 1c, Extended Data Fig. 2b**) to D_E of 10 and 20s (**Extended Data Fig. 2c-d**). To prevent the confusion between these graphs, we revised the manuscript by removing **Extended Data Fig. 2a, b, e, f**. Also, with this information in background, we provided explanations regarding the questions as below.

1) Effect of A-IDP process on orientation of the quasi-2D phases

The orientation of the quasi-2D phase perovskite crystals can be tuned by IDP process. First, the pristine film exhibits orientation of the low- n ($n = 1 - 3$) phases toward the out-of-plane (Q_z) direction (**Fig. R2a**). With IDP process with D_E of 0s, THF provides dissolution of intermediate complex of PEA while reducing the solubility of 3D CsPbBr_3 precursors at the same time, facilitating crystallization of middle- n phases even before the annealing process (**Extended Data Fig. 1d**). With increased D_E , the initially crystallized middle- n phases act as seed that induces heterogeneous nucleation (crystal growth) to be the dominant process in an energetically favorable manner thus enhance the crystallinity, rather than the homogeneous nucleation of randomly oriented individual crystals (*Nano Lett.* 24, 17, 5308–5316 (2024)). This results in increased crystallinity of medium- n phases, making the Q_z direction of Bragg spots at 1.07 \AA^{-1} to be stronger. We revised the manuscript with explanation regarding the effect of A-IDP process on the crystal orientation as below.

Revised parts in the manuscript

In the page 8, line 21

“In contrast, perovskite film prepared by A-IDP showed only ring-patterned scattering peaks, which correspond to 3D CsPbBr₃ (5.8 Å) oriented **mainly toward the Q_z direction, indicating increased crystallinity of medium-*n* phases toward vertical direction that can benefit on efficient charge transport in LED devices.**”

Fig. R2 | Effect of IDP process on crystal structure of quasi-2D perovskite film. GIWAXS profile of perovskite thin film without IDP process (a), after THF solution exposure time of 0 s (S-IDP) (b), after CCA-THF solution exposure time of 0s (A-IDP) (c), 10 s (d), and 20 s (e).

2) Effect of IDP process on additional phase at 0.9 Å⁻¹

The diffraction peak from Bragg spot for 0.9 Å⁻¹ can be assigned to (200) direction diffraction peak of isolated trigonal 0D Cs₄PbBr₆ phase (*J. Phys. Chem. Lett.* 2017, 8, 3, 565–570), from excess stoichiometric ratio of A-site cations that we used in precursors; PEABr, CsBr, (EABr_{1-x}: (MDACl₂)_x), and PbBr₂ in molar ratio of 2: 1.6 : 2.4 : 3 in DMSO. The presence of the 0D Cs₄PbBr₆ phase could be also verified from steady-state UV-Vis absorption spectra, showing absorption peak at 320 nm especially in pristine perovskite film (**Fig. R3**). The excess stoichiometric ratio of A-site cation was incorporated to prevent segregation of metallic Pb clusters from any Pb-excess stoichiometry (*Science*, 350(6265), 1222-1225, 2015) and

segregation of low- n phases (*Nanoscale*, 11, 3546-3556 (2019)). Here, the IDP process with increasing D_E results in removal of the intermediate complex including PEA, which also enabled excess CsBr molecules to be easily bind with $\text{Pb}(\text{Br}/\text{Cl})_x^-$ octahedral to make 3D cubic CsPbBr_3 phase, rather than making the insulating CsBr-rich Cs_4PbBr_6 phases.

Fig. R3 | Normalized UV-vis absorption and PLE spectra of pristine and A-IDP perovskite films. The PLE spectra was normalized at the band edge with respect to the absorption spectra.

Q4. The authors mentioned that in pristine and A-IDP films, the ground state bleaching (GSB) peaks of different n -value phases were probed at same locations (Fig. 3e-f). It seems contradictory that the CCA can intercalate into the lattice of quasi-2D perovskites (Fig. 1). As well documented, the mixed spacer cations would change the bandgap of quantum wells (Nat. Commun. 2018, 9, 3541; Adv. Mater. 2021, 2104381), leading to the slight shifting of GSB peaks.

Response) Thank you for the valuable comment. As the reviewer mentioned, the bandgap of each n phase in quasi-2D perovskite structures can change due to variations in the Br–Pb–Br distance and lattice distortions arising from different interactions between the organic spacer and the octahedral lattice (Adv. Mater. 36 (11), 2404517 (2024)). Many previously reported quasi-2D perovskite systems with large bandgap shifts used organic cations such as isopropylammonium⁺ (IPA⁺) or methylbenzylammonium⁺ (MBZA⁺), which induce significant changes in the d -spacing and involve more than 25% cation substitution (Nat. Commun. 9, 3541 (2018); Adv. Mater. 2104381 (2021)). In contrast, in our study, partial substitution of PEA in $n = 1$ PEA₂PbBr₄ with CCA (Fig. 1d) changed the d -spacing from 16.6 Å to 16.7 Å, only about 0.1 Å. This modest variation suggests that any resulting alteration in the electronic structure or bandgap is minimal, given the similarly small Br–Pb–Br distance and degree of lattice distortion. Similarly, the bandgap of (5-AVA)₂PbBr₄ (5-AVA: 5-aminovaleric acid), which has a similar lattice d -spacing of 16.2 Å comparable to that of PEA₂PbBr₄, was reported to be similar band gap of ≈ 3.02 eV (see Eur. J. Inorg. Chem. 2020, 4581–4592).

Also, the second-derivative-based deconvolution of our UV–Vis absorption and transient absorption spectra shows a slight redshift in the bandgap of the low- n ($n = 1, 2$) phase in A-IDP (Fig. R4). Although this indicates a subtle difference compared to the pristine sample, the overall bandgap shift is minor, which can be attributed to the minimal lattice changes introduced by CCA.

Fig. R4 | Deconvoluted second-derivative of UV-Vis spectra (a, b) and corresponding bandgap of pristine and A-IDP perovskite.

Q5. The turn-on voltage of S-IDP and A-IDP PeLEDs were both achieved at 2.2 V, which were quite lower than the bandgap ($E_g = \sim 2.54$ eV). The corresponding device simulations and sub-bandgap turn-on mechanism should be discussed.

Response) We sincerely appreciate the reviewer's comment regarding the sub-bandgap turn-on behavior in our S-IDP and A-IDP PeLEDs. As described, both devices exhibit a turn-on voltage at 2.2 V, which is notably lower than their bandgap of approximately 2.54 eV. This sub-bandgap turn-on can be explained by non-thermal-equilibrium band-edge carriers whose populations are determined by the Fermi-Dirac function perturbed by a small external bias, where even a small bias can facilitate radiative recombination in the perovskite emissive layer (*Nat. Commun.* 2022, 13, 3845). Under thermal equilibrium, a population of carriers near the band edges already exists; even with small external voltage lower than the E_g/q is applied, the partial cancellation of built-in potential can facilitate these pre-existing carriers to undergo radiative recombination. Recent work also experimentally provided that increasing the concentration of minor carriers (hole in this case, with high electron injection affinity of PO-T2T as an ETL) with high carrier density of major carriers can trigger measurable light emission at biases lower than the optical bandgap, as the energetic deficit is offset by the non-equilibrium distribution of carriers (**Fig. R5**) (*Nat. Mater.* (2025) doi: 10.1038/s41563-025-02123-y). Here, the effect of ETL for distribution of electron as major carrier was also confirmed, showing PeLEDs with only PO-T2T as ETL showed sub-gap turn-on voltage, which was not shown by PeLEDs with TPBi as ETL (**Fig. R5**).

In case of our devices, S-IDP and A-IDP PeLEDs could provide the uniform energy distribution without segregated low- n phases can facilitate charge transfer and radiative recombination as indicated in the transient EL analysis (**Extended Data Fig. 8a-c**). Also, combined with our device structure with high electron transport affinity of CN-T2T as ETL, sub-gap turn-on behavior could be observed as like the reports elsewhere (**Table R1**). We provided additional discussion in manuscript as below to indicate the sub-gap turn-on mechanism as below.

Revised parts in the manuscript

In the page 13, line 30

“Compared to pristine PeLEDs, which exhibit a turn-on voltage greater than 3.0 V, both S-IDP and A-IDP PeLEDs showed a sub-gap turn-on voltage of 2.2 V—below the bandgap ($E_g = 2.54$ eV). This observation can be attributed to the recombination of non-thermal-equilibrium

band-edge carriers in the emitting layer, where the carrier populations are perturbed by a small external bias, leading to radiative recombination below the turn-on voltage^{38,39}. Furthermore, PeLEDs processed via the IDP method benefited from improved charge transport and recombination of band-edge carriers, which can be explained by the suppression of low-n phases and negligible charge accumulation.”

[FIGURE REDACTED]

Fig. R5 | J-V-L curves of different PeLEDs. **a**, ITO/PMMA/PTAA/Perovskite/PO-T2T/LiF/Al, **b**, ITO/PTAA/Perovskite/PO-T2T/LiF/Al, **c**, ITO/TFB:PVK(4:6)/Perovskite/PO-T2T/LiF/Al, for PO-T2T based devices and **d**, ITO/PMMA/PTAA/Perovskite/TPBi/LiF/Al, **e**, ITO/TFB/PTAA/Perovskite/TPBi/LiF/Al, **f**, ITO/TFB:PVK(4:6)/Perovskite/TPBi/LiF/Al, for TPBi based devices. The figures are adapted from Qin, J. et al. Nat. Mater. 2025, doi: 10.1038/s41563-025-02123-y.

Table R1 | Summarized electrical characteristics of reported PeLEDs.
ETL: Electron transport layer, $V_{\text{turn-on}}$: Voltage at luminance of 1cd m⁻².

Device Structure	ETL	Mobility of ETL (μ) (cm ² V ⁻¹ s ⁻¹)	Bandgap (eV)	$V_{\text{turn-on}}$ (V)	Ref.
ITO/TFB/LiF/EML/ PO-T2T/LiF/Al	PO-T2T	4.4×10^{-3}	2.40	2.0	Nat. Commun. 13:3845 (2022)
ITO/PMMA/PTAA/EML/ TPBi/LiF/Al	TPBi	2.2×10^{-5}	2.38	2.6	Nat. Mater. (2025). doi:10.1038/s41563- 025-02123-y
ITO/PMMA/PTAA/EML/ ZADN/LiF/Al	ZADN	2.2×10^{-4}	2.38	2.2	
ITO/PMMA/PTAA/EML/ PO-T2T/LiF/Al	PO-T2T	4.4×10^{-3}	2.38	2.0	
ITO/Buf-HIL/EML/ CN-T2T/LiF/Al	CN-T2T	$10^{-4} - 10^{-5}$	2.54	2.22	This work

Q6. The related performances and efficiency calculations of PeLEDs by placing on the light-outcoupling hemispherical lens should be provided. Whether the emission still

follow Lambert's cosine law?

Response) Thank you for the critical comment. To comply with the reviewer's comment, we provided the efficiency calculation process and further verified it through measurements based on both integrated sphere and spectroradiometer setup.

First, the total external quantum efficiency (EQE), η_{EQE} of device can be calculated as follows:

$$\eta_{EQE} = \frac{\# \text{ of externally ejected photons toward upp. hemisphere}}{\# \text{ of injected carriers}} \quad \dots (1)$$

$$= \frac{\int \Psi_R^{(u.hemi)}(\lambda)/(hc/\lambda)d\lambda}{i_{LED}/e} = \frac{e}{hc} \frac{\int \lambda \Psi_R^{(u.hem)}(\lambda)d\lambda}{i_{LED}} \quad \dots (2)$$

$$= \frac{e}{hc} \frac{\int_{\lambda} \int_{u.hemi} \lambda L_R(\theta, \phi, \lambda) A_S \cos\theta d\Omega d\lambda}{i_{LED}} \quad \dots (3)$$

$$= \frac{e}{hc} \frac{\int_{\lambda} \int_{\phi=0}^{2\pi} \int_{\theta=0}^{\pi/2} \lambda L_R(\theta) \cos\theta \sin\theta d\theta d\phi d\lambda}{J_{LED}} \quad \dots (4)$$

where $\Psi_R^{(u.hemi)}(\lambda)$ is the total emission power, i_{LED} is current, $L_R(\theta, \phi, \lambda)$ is the radiance, and A_S is active area.

From Eqn. 2, we can calculate the relative ratio factor of η_{EQE} with or without the hemisphere lens at same current density as below,

$$r = \frac{\eta_{EQE \text{ with hemisphere}}}{\eta_{EQE \text{ witho hemisphere}}} = \frac{\Psi_R^{(u.hemi \text{ with hemisphere})}(\lambda)}{\Psi_R^{(u.hemi \text{ without hemisphere})}(\lambda)} \quad \dots (5)$$

where each $\Psi_R^{(u.hemi)}(\lambda)$ can be measured by integrating sphere that allows measuring the emission power of devices in total angle (*Nat. Photon.* 562, 249–253 (2018)) (**Fig. R5a**). Under the same conditions used for the actual J-V-L measurements, we compared the integrated EL intensity measured inside the integrating sphere before and after attaching the hemisphere lens and observed a ratio of $r \approx 1.85$ at each voltage (**Fig. R5b**). From this ratio, the calculated maximum EQE increased from 22.5% (without hemisphere) to 41.8% (with hemisphere). Such an increase in outcoupling efficiency upon attaching the hemisphere results from a reduction in the internal total reflection (i.e., a lower critical angle), which decreases the proportion of waveguided modes that propagate laterally and thereby boosts outcoupling efficiency in the

forward (normal) direction compared to a Lambertian distribution (*J. Display Technol.* 2 (2), 143-152 (2006)).

[FIGURE REDACTED]

Figure R5 | Outcoupling efficiency measurement setup of perovskite LEDs with hemisphere lens. **a**, Schematic illustration of measurement setup using an integrating sphere and a spectrometer. *The schematic is adapted from Nature Photonics 562, 249–253 (2018).* **b**, Relative EL intensity versus voltage of perovskite LEDs measured in the integrating sphere and the ratio of EL intensity depending on use of hemisphere lens.

To comply with the reviewer's comment and verify the calculated EQE based on the integrating sphere setup, we also verified this effect by measuring the angle-dependent emission profile before and after attachment of hemisphere lens (**Fig. R6a**). When calculating the EQE based on the emission profile as a function of viewing angle of θ , the process follows Equation (2) and includes the additional steps described below;

$$\frac{e}{hc} \frac{2\pi L_{R0}}{J_{LED}} \int_{\lambda_i}^{\lambda_f} \lambda \rho(\lambda) d\lambda \int_{\theta=0}^{\pi/2} g(\theta) \cos\theta \sin\theta d\theta \quad \dots (6)$$

$$= \eta_{EQE}^{(lambertian)} \int_{\theta=0}^{\pi/2} 2g(\theta) \cos\theta \sin\theta d\theta \quad \dots (7)$$

$$= f \times \eta_{EQE}^{(lambertian)} \quad \dots (8)$$

where

$$g(\theta) = \frac{L_R(\theta)}{L_{R0}} (= 1 \text{ if Lambertian}) \quad \dots (9)$$

and

... (10)

$$f = \int_{\theta=0}^{\pi/2} 2g(\theta)\cos\theta\sin\theta d\theta$$

Here, the f -factor can be used to calculate the EQE with different angle-dependent emission profile by combining Eqn. 5 and Eqn. 8 as below.

$$r = \frac{\eta_{EQE \text{ with hemisphere}}}{\eta_{EQE \text{ without hemisphere}}} = \frac{f_{\text{with hemisphere}}}{f_{\text{without hemisphere}}} \quad \dots (11)$$

Similar with the angle-dependent profile reported elsewhere with hemisphere lens, relative emission intensity greatly increased at region of low angle (forward direction) and resulted in in $r = 1.87$, similar with that obtained from the measurement based on integrating sphere (**Fig. R6b**).

Figure R6 | Angle-dependent measurement of perovskite LEDs. **a**, Schematic illustration of measurement setup for angle-dependent emission of perovskite LEDs with goniometric setup. **b**, Angle-dependent EL intensity of perovskite LEDs depending on use of hemisphere lens. The f -factors are calculated by integrating the area of angle-dependent EL profile on polar coordinates.